# The Use of CHROMID^®^ Colistin R for the Detection of Colistin-Resistant Gram-Negative Bacteria in Positive Blood Cultures

**DOI:** 10.3390/antibiotics13030246

**Published:** 2024-03-07

**Authors:** Emma C. L. Marrs, Olivia Milburn, Gary J. Eltringham, Danielle J. C. Fenwick, Sylvain Orenga, Gulsen Hazırolan, Pinar Zarakolu, John D. Perry

**Affiliations:** 1Microbiology Department, Freeman Hospital, Newcastle upon Tyne NE7 7DN, UK; e.marrs@nhs.net (E.C.L.M.); livvy@fxpilates.com (O.M.); gary.eltringham@nhs.net (G.J.E.); danielle.fenwick@nhs.net (D.J.C.F.); 2Research & Development Microbiology, bioMérieux SA, 38 390 La Balme-les-Grottes, France; sylvain.orenga@biomerieux.com; 3Department of Medical Microbiology, Faculty of Medicine, Hacettepe University, Ankara 06230, Türkiye; gcetin@hacettepe.edu.tr; 4Department of Infectious Diseases and Clinical Microbiology, Faculty of Medicine, Hacettepe University, Ankara 06230, Türkiye; zarakolu@hacettepe.edu.tr

**Keywords:** colistin, chromogenic medium, antibiotic resistance, blood culture

## Abstract

The aim of this study was to assess the utility of CHROMID^®^ Colistin R for direct detection of colistin-resistant Gram-negative bacteria from positive blood cultures. A total of 390 blood cultures from hospitalised patients containing Gram-negative bacteria were included in this study. These blood cultures were referred to clinical laboratories in the United Kingdom and Türkiye. A further 16 simulated positive blood culture bottles were included that contained a range of colistin-resistant strains as well as susceptible control strains. Fluid from each positive blood culture was diluted 1/200 in saline and 10 µL aliquots cultured onto cystine-lactose-electrolyte-deficient agar and CHROMID^®^ Colistin R. All recovered bacteria were identified, and for Gram-negative bacteria, their minimum inhibitory concentration of colistin was measured using the broth microdilution method. From a total of 443 Gram-negative isolates, 57 colistin-resistant isolates were recovered, of which 53 (93%) grew on CHROMID^®^ Colistin R within 18 h. Of the 377 isolates determined to be colistin-susceptible, only 9 isolates were able to grow, including 6 isolates of *Pseudomonas aeruginosa*. For positive blood cultures that are shown to contain Gram-negative bacteria, culture on CHROMID^®^ Colistin R is a useful diagnostic tool to detect susceptibility or resistance to colistin within 18 h.

## 1. Introduction

In recent years, colistin has been increasingly used as a last resort treatment for multidrug-resistant (MDR) Gram-negative bacteria causing serious infections such as bacteraemia or hospital-acquired pneumonia [1,2,3,4,5]. Treatment with colistin commonly results in nephrotoxicity and, especially as monotherapy, is associated with poor clinical outcomes [6]. Despite this, usage is driven by the fact that colistin may be the only drug showing in vitro activity against some MDR Gram-negative bacteria [7], and treatment options may be further limited in low-income countries where newer antibiotics may be unavailable and/or unaffordable [8,9]. For these reasons, and despite well-documented reservations over its efficacy, the European Committee on Antimicrobial Susceptibility Testing (EUCAST) has retained breakpoints for the interpretation of colistin susceptibility for Enterobacterales, *Acinetobacter* species and *Pseudomonas* species [6]. However, such species should only be reported as “susceptible” to colistin if it is supported by “other active therapy” [10].

The use of colistin (and other polymyxins) is further complicated by difficulties associated with antimicrobial susceptibility testing (AST) in the clinical laboratory. For example, the use of disc diffusion and gradient strips that both rely on agar diffusion is known to be unreliable for testing colistin [11,12]. Consequently, only broth microdilution is recommended by EUCAST [13]. This precludes the testing of colistin as part of EUCAST’s rapid AST method for direct testing of Gram-negative bacteria in positive blood cultures, as the method relies on disc diffusion [14].

For patients with sepsis, it is well established that survival is directly linked to the prompt initiation of effective antibiotic therapy [15,16]. It would therefore be desirable to identify a reliable method for direct testing of positive blood cultures that can establish susceptibility or resistance to colistin. This would allow colistin to be considered (or excluded) as a treatment option for cases of bacteraemia caused by MDR Gram-negative bacteria where few other options are available.

CHROMID^®^ Colistin R is a recently developed chromogenic culture medium that allows the growth of colistin-resistant Gram-negative bacteria as coloured colonies [17]. The objective of this study was to determine if plating positive blood cultures onto CHROMID^®^ Colistin R would facilitate detection of colistin resistance within 18 h and preclude the need for further testing by broth microdilution.

## 2. Results

### 2.1. Recovery of Various Strains from Simulated Blood Cultures

All strains recovered from 16 simulated blood cultures grew well on CLED agar, generating at least 100 colonies. Three Gram-positive strains, *Staphylococcus aureus* ATCC 29213, *Enterococcus faecalis* ATCC 29212 and *Candida albicans* ATCC 90028, were completely inhibited by CHROMID^®^ Colistin R, and the same was true of two Gram-negative control strains known to be susceptible to colistin (*E. coli* NCTC 12241 and *P. aeruginosa* NCTC 12903) with MICs of 0.5 and 1 mg/L, respectively. The other eleven strains all formed colonies on CHROMID^®^ Colistin R, although one resistant strain of *Enterobacter cloacae* (MIC = 16 mg/L) with *mcr-1* was only able to generate a single green colony (see Figure 1). All eleven strains were resistant to colistin, except for one strain of *P. aeruginosa* (MIC = 4 mg/L).

### 2.2. Results with Blood Cultures from Patients in Newcastle upon Tyne, UK

Any samples that did not yield colonies of Gram-negative bacteria on either CLED agar or CHROMID^®^ Colistin R were excluded from this study; this resulted in the exclusion of non-target fastidious bacteria (e.g., *Haemophilus influenzae*) and anaerobes (e.g., *Bacteroides* species). A total of 326 positive BACT-ALERT^®^ blood culture bottles were processed and included in this study. These included 157 aerobic bottles, 137 anaerobic bottles and 32 paediatric bottles. A total of 297 bottles contained a single Gram-negative species, and 29 bottles had a polymicrobial culture that included at least one Gram-negative species. A total of 382 isolates were recovered, including 364 Gram-negative isolates (see Table 1) and 18 Gram-positive isolates. All of the eighteen Gram-positive isolates were recovered on CLED, but only one of the isolates, a vancomycin-resistant strain of *Enterococcus faecium*, was recovered on CHROMID^®^ Colistin R. The remainder were inhibited, including *Bacillus* species (*n* = 2), *Candida* species (*n* = 1), *Staphylococcus epidermidis* (*n* = 2), *Streptococcus* species (*n* = 3) and *Enterococcus* species (*n* = 9).

### 2.3. Results with Blood Cultures from Patients in Hacettepe University Hospitals, Ankara, Türkiye

Sixty-four blood cultures from patients in Hacettepe University Hospitals, Ankara, were included in this study, and these yielded a total of 66 Gram-negative isolates (see Table 1). Only two Gram-positive strains were recovered on CLED, and they both failed to grow on CHROMID^®^ Colistin R.

### 2.4. Ability of CHROMID^®^ Colistin R to Detect Colistin-Resistant Gram-Negative Bacteria

Susceptibility to colistin was established by performing BMD on all 443 isolates of Gram-negative bacteria recovered in the UK (*n* = 364) and Türkiye (*n* = 66) and the Gram-negative strains (*n* = 13) used to create simulated blood cultures. Table 2 shows all of the colistin-resistant strains used or found in this study and whether or not they were recovered on CHROMID^®^ Colistin R.

A total of 57 isolates of Gram-negative bacteria were resistant to colistin, and 53 of these were recovered on CHROMID^®^ Colistin R (sensitivity: 93%). Thirty-four isolates represented species that are normally intrinsically resistant to colistin (i.e., *Morganella*, *Proteus* and *Serratia*). The remaining 23 isolates represented species that are normally susceptible to colistin but where colistin resistance has been acquired. Eleven of these were from simulated blood cultures, and twelve were from real blood cultures. The proportion of Gram-negative isolates from real blood cultures with acquired resistance to colistin (among species without intrinsic resistance) was significantly higher in Türkiye when compared to the UK (10.3% versus 1.5%; *p* = 0.001), although the study in Türkiye was performed several months after the UK study.

Colistin-resistant strains from intrinsically resistant species all grew on CHROMID^®^ Colistin R. Among the 23 isolates with acquired resistance to colistin, 19 were recovered on CHROMID^®^ Colistin R (83%). Among the Enterobacterales, two isolates of *E. cloacae* (colistin MIC > 8 mg/L) from two patients in the UK failed to grow on CHROMID^®^ Colistin R, as did one isolate of *K. pneumoniae* (colistin MIC > 8 mg/L) from a patient in Türkiye. One colistin-resistant *Pseudomonas putida* (MIC > 8 mg/L) from a UK patient also failed to grow on CHROMID^®^ Colistin R. The transmissible resistance gene *mcr-1* was known to be produced by two of the isolates used in the inoculation of simulated blood cultures, including *E. coli* NCTC 13846 and an isolate of *E. cloacae*. Both produced positive results in the PCR assay for *mcr* genes and were recovered on CHROMID^®^ Colistin R. None of the other isolates from this study with acquired resistance to colistin harboured genes for *mcr 1–3*.

### 2.5. Ability of CHROMID^®^ Colistin R to Predict Colistin Resistance among Enterobacterales

According to the instructions for the use of CHROMID^®^ Colistin R in medical microbiology, the medium is designed to target Enterobacterales, including *E. coli*, *K. pneumoniae*, *Salmonella* spp and *Enterobacter* spp. An analysis was therefore performed to examine its performance in detecting colistin resistance, specifically among Enterobacterales. The data are shown in Table 3.

Table 3 shows that for Enterobacterales, only 3 out of 327 colistin-susceptible isolates were able to grow on CHROMID^®^ Colistin R, and 51 of 54 resistant isolates were successfully detected. For the detection of Enterobacterales, sensitivity and positive predictive value were 94.4%, whereas specificity and negative predictive value were 99.1% (Table 4).

### 2.6. False-Positive Isolates Recovered on CHROMID^®^ Colistin R

Nine colistin-susceptible strains were recovered on CHROMID^®^ Colistin R from the 406 blood cultures processed in the UK and Türkiye. These isolates included three Enterobacterales (two *K. pneumoniae* and *E. coli*) that formed only 1–2 colonies. All three isolates had a colistin MIC of 0.5 mg/L. In addition, six isolates of *P. aeruginosa* (with colistin MICs 1–4 mg/L) were recovered on CHROMID^®^ Colistin R, and each generated good growth (>100 colonies).

Finally, four Gram-negative strains were recovered on CHROMID^®^ Colistin R, for which interpretive susceptibility criteria are not available. These included a *Sphingobacterium* species (colistin MIC > 8 mg/L) and three isolates of *Stenotrophomonas maltophilia* (MICs were 4, >8 and >8 mg/L).

### 2.7. Summary of Performance of CHROMID^®^ Colistin R

For the 443 Gram-negative isolates recovered from blood cultures, EUCAST breakpoints were applicable to 434 isolates (i.e., all Enterobacterales, *Pseudomonas* species and *Acinetobacter* species)—thus allowing categorisation as susceptible or resistant. Table 4 summarizes the performance of CHROMID^®^ Colistin R based on the premise that recovery of any colonies of a particular Gram-negative strain on this medium indicates resistance to colistin and lack of growth indicates susceptibility. A “positive result” is deemed as the growth (and therefore resistance) of Enterobacterales, *Acinetobacter* species or *Pseudomonas* species. Table 4 also shows the same analysis when only Enterobacterales are considered.

## 3. Discussion

EUCAST has developed a rapid antimicrobial susceptibility testing method for direct testing of positive blood cultures that can generate results for Enterobacterales in as little as 4 h [14,18]. This is extremely useful to quickly assess whether patients with bacteraemia or sepsis are receiving optimal antimicrobial therapy. The test is based on disc diffusion and is therefore unsuitable for testing colistin. A variety of methods have been developed for rapid determination of colistin susceptibility, and these were recently reviewed by Leshaba et al. [19]. Several agar-based methods are available, including CHROMagar™ Col-APSE [20], SuperPolymyxin [21] and CHROMID^®^ Colistin R [17], but few methods have been applied to direct testing of positive blood cultures. Malli et al. applied the Rapid Polymyxin™ NP test, which is a 2–3 h test based on the acidification of glucose in the presence of polymyxin, to the detection of colistin-resistant Enterobacterales directly from positive blood cultures [22]. They reported a sensitivity and negative predictive value of 100% and a positive predictive value of 76.5%. From 132 blood cultures, nine isolates were found to have acquired resistance to colistin, and there were four false-positive results. Fonseca et al. described a flow cytometry assay and applied it to simulated blood cultures spiked with 204 Gram-negative bacilli (137 Enterobacterales, 35 *Pseudomonas* spp. and 32 *Acinetobacter baumannii*) [14]. Bacteria were extracted from positive blood cultures and incubated for 1 h at 37 °C with 2 and 4 mg/L colistin and a fluorescent dye. Cells were then analysed on two different flow cytometers. The overall assay time was 2 h, and the authors reported a 99% categorical agreement with MICs derived from broth microdilution [23].

Culture of positive blood cultures onto CHROMID^®^ Colistin R requires 18 h incubation but offers an extremely simple methodology and requires no specialised equipment. Among the Enterobacterales, the value of such an assay depends on its ability to detect strains with acquired colistin resistance, and 17 out of 20 such strains were reliably detected in this study. The medium failed to recover two isolates of the *E. cloacae* group, and a weakness in the detection of *Enterobacter* spp. on CHROMID^®^ Colistin R has been previously noted. For example, CHROMID^®^ Colistin R was used to screen for colistin-resistant Enterobacterales from stool samples and rectal swabs. The authors reported an overall sensitivity of 88.1% but a low sensitivity of 57.1% for the detection of *Enterobacter* species [17]. Bala et al. investigated the use of CHROMID^®^ Colistin R for detection of resistance in 87 well-characterized Enterobacterales using an inoculum of 10^4^ CFU/plate. They concluded that CHROMID^®^ Colistin R provides a safe, reliable and convenient alternative to BMD for colistin antimicrobial susceptibility testing in *Escherichia coli*, *Klebsiella pneumoniae* and *Salmonella* spp., but for *Enterobacter* spp., inferring susceptibility was unreliable (even when MICs were >64 mg/L), and further improvements were required [24]. The authors speculated that difficulties in detecting *Enterobacter* may be related to the phenomenon of heteroresistance in some strains of this species [25]. Heteroresistance in *Enterobacter* has also proven to be difficult to detect using other methods, including commercial systems, e.g., Vitek 2 [26]. Finally, the instructions for the use of CHROMID^®^ Colistin R state that some strains of *Enterobacter* resistant to colistin may not grow on the medium, and in the case of a negative result and a strong suspicion of colistin-resistant carriage, an alternate method should be used.

## 4. Methods

### 4.1. Preparation and Processing of Simulated Blood Cultures

Sixteen microbial strains were subcultured onto Columbia agar (Oxoid, Basingstoke, UK) plus 5% horse blood and incubated for 24 h at 37 °C. The strains comprised thirteen Gram-negative bacteria with known MICs to colistin ranging from 0.5 to 64 mg/L, two Gram-positive bacteria and one isolate of *Candida albicans* (strain details are provided in Section 2.1 and Figure 1). Strains designated as ATCC were obtained from the American Type Culture Collection, Manassas, VA, USA, and those designated as NCTC were obtained from the National Collection of Type Cultures, Colindale, UK. The other strains were derived from the culture collections of the Freeman Hospital Microbiology Department, Newcastle upon Tyne, UK, and bioMérieux, La Balme-les-Grottes, France. Each isolate was suspended in sterile saline (0.85%) to a turbidity of 0.5 McFarland units using a Densimat (bioMérieux). A 10 µL aliquot of this suspension was then added to 10 mL of sterile saline to produce an inoculum of approximately 100,000 CFU/mL. A 10 µL aliquot of each diluted suspension was then added to 10 mL of sterile defibrinated horse blood to produce an inoculum of approximately 100 CFU/mL (note: for *Candida albicans*, 300 µL was used instead of 10 µL to produce an equivalent inoculum). Inoculated horse blood (10 mL) was then added to an aerobic BACT-ALERT^®^ blood culture bottle and loaded onto a BACT/ALERT^®^ 3D automated microbial detection system (bioMérieux, Saint Louis, MO, USA).

By the following day, all bottles had signalled positive on the BACT-ALERT^®^ instrument, at which point each bottle was mixed by inverting three times, and approximately 200 µL was removed using a needle and added to a sterile plastic bottle. A 5 µL aliquot of this fluid was then added to 10 mL of sterile saline and vortexed for 5 s. A 10 µL aliquot of this dilution was then added to a plate of CHROMID^®^ Colistin R (Ref: 421170; bioMérieux, Basingstoke, UK) and a plate of cystine-lactose-electrolyte-deficient agar (CLED; Oxoid, Basingstoke, UK). These inocula were spread to obtain isolated colonies, and the plates were incubated at 37 °C in air for 18 h. For each strain, the growth and colouration of any recovered colonies were recorded for CHROMID^®^ Colistin R. Growth on CLED was also recorded, and the culture was checked for purity.

### 4.2. Processing of Blood Cultures from Patients in Newcastle upon Tyne, UK

For any routine blood culture sample that was signalled as positive by the BACT/ALERT^®^ instrument, the bottle was mixed and fluid removed for Gram stain. For blood cultures that were positive during the day (8 a.m.–5 p.m.), the bacteria in the blood culture sample were concentrated by centrifugation after treatment with saponin, and species identification was attempted using matrix-assisted laser desorption ionization–time of flight mass spectrometry (MALDI-TOF MS; Bruker, Coventry, UK). Any positive blood culture containing Gram-negative bacteria (based on Gram stain and/or MALDI TOF MS result) was included in this study, and a 200 µL aliquot was removed and anonymized with a study number. The bottle type (e.g., aerobic, anaerobic, paediatric) was recorded, as were any results of bacterial identification. Duplicate blood cultures from the same patient within 10 days of each other were excluded. A 5 µL aliquot was diluted and cultured on CHROMID^®^ Colistin R and CLED exactly as described in Section 4.1.

### 4.3. Processing of Blood Cultures from Patients in Hacettepe University Hospitals, Ankara, Türkiye

In Ankara, the BD Bactec^TM^ FX instrument (Becton Dickinson, Istanbul, Türkiye) was used for routine monitoring of blood cultures. Any positive samples shown to contain Gram-negative bacteria by Gram stain were cultured onto CHROMID^®^ Colistin R and CLED medium exactly as described in Section 4.2. Duplicate blood cultures from the same patient within 10 days of each other were excluded.

### 4.4. Interpretation of Culture Plates

The same protocol was used in both laboratories. The colony appearance and number of colonies for each colony type recovered on CLED and/or CHROMID^®^ Colistin R were recorded, and each colony type was stored for further testing. Species identification was confirmed on all colony types using MALDI-TOF MS (Bruker) in both laboratories. An absence of growth on both of the media correlated with the presence of fastidious species (e.g., *Haemophilus influenzae*) or strict anaerobes (e.g., *Bacteroides* species), and these blood cultures were excluded from this study.

### 4.5. Determination of the Minimum Inhibitory Concentration (MIC) of Colistin

All Gram-negative isolates recovered on both CLED and CHROMID^®^ Colistin R were subcultured onto Columbia blood agar for 18 h to obtain a fresh pure subculture, and they were tested for susceptibility to colistin using the broth microdilution method in strict accordance with the Clinical and Laboratory Standards Institute (CLSI) methodology [27]. Colistin sulphate (CAS No.: 1264-72-8; Discovery Fine Chemicals, Wimborne, UK) was used for testing, and, after adjusting for potency, the final concentrations of ‘active’ colistin ranged from 8 to 0.125 mg/L. Antibiotic-free wells were included for each isolate tested, as well as sterility controls without any bacteria. Control strains of known MICs were also tested, including *Escherichia coli* NCTC 12241, *E. coli* NCTC 13846 and *Pseudomonas aeruginosa* NCTC 12903. Strains were interpreted as susceptible or resistant based on EUCAST breakpoints, with resistance being defined as a colistin MIC > 2 mg/L (for Enterobacterales and *Acinetobacter* species) or >4 mg/L (for *Pseudomonas* species) [13].

### 4.6. Determination of mcr Genes Using Polymerase Chain Reaction (PCR)

Any Gram-negative isolates that were recovered on CHROMID^®^ Colistin R or proved to be colistin-resistant by MIC testing were tested by PCR for the presence of *mcr* genes encoding transmissible colistin resistance. Exceptions included any isolates of *Proteus*, *Morganella* or *Serratia*, which are known to be intrinsically resistant to colistin. Testing was based on the method of Li et al. [28], allowing detection of *mcr-1*, *mcr-2* and *mcr-3*. Each isolate to be tested (including *E. coli* NCTC 13846 and *E. coli* NCTC 12241 as positive and negative controls, respectively) was subcultured onto Columbia blood agar for 18 h. A suspension equivalent to 5.0 McFarland units was prepared using a Densimat, and two dilutions (1/100 and 1/1000) were prepared from each suspension. The diluted samples underwent extraction and purification using ROCHE MagNAPure 96 DNA and Viral NA Small Volume Kit (REF (240)06543588001; Roche Diagnostics Ltd., Burgess Hill, UK), extracting from a 200 µL volume of sample input into a 100 µL final eluate. A 5.0 µL aliquot of each eluate was used as a template in the PCR reaction along with 10.0 µL Fast SYBR^®^ Green Master Mix (Applied Biosystems 4385610; Thermo Fisher Scientific, Horsham, UK), 4.2 µL of molecular-grade water and 0.4 µL of each primer (forward primer AAAGACGCGGTACAAGCAAC and reverse primer GCTGAACATACACGGCACAG) to give a final concentration of 0.2 µM in the final reaction volume.

The assay was performed on the Applied Biosystems FAST 7500 real-time PCR platform (Thermo Fisher Scientific, Horsham, UK) employing the following cycling conditions: a denaturation step of 95 °C for 20 s, followed by 40 cycles of 95 °C for 3 s, and 60 °C for 30 s. A dissociation step was added following amplification, allowing melt curve analysis of any fluorescent signal generated to be differentiated between target amplicon and primer–dimer formation based on a positive melting curve of around 85 °C.

### 4.7. Statistical Analysis

Fisher’s exact test was used to compare the proportion of Gram-negative isolates resistant to colistin among blood culture isolates in the United Kingdom and Türkiye.

## 5. Conclusions

Inoculation of positive blood cultures containing Gram-negative bacilli onto CHROMID^®^ Colistin R is a very straightforward procedure that can provide an early indication of likely resistance to colistin. For Enterobacterales, the growth of >10 colonies always correlated with resistance, as confirmed by BMD. If colistin therapy is under consideration for strains that appear susceptible, confirmation of the susceptible result using BMD is advisable, particularly for strains of *Enterobacter* species, at least until further studies are able to confirm the accuracy of this method.

## Figures and Tables

**Figure 1 antibiotics-13-00246-f001:**
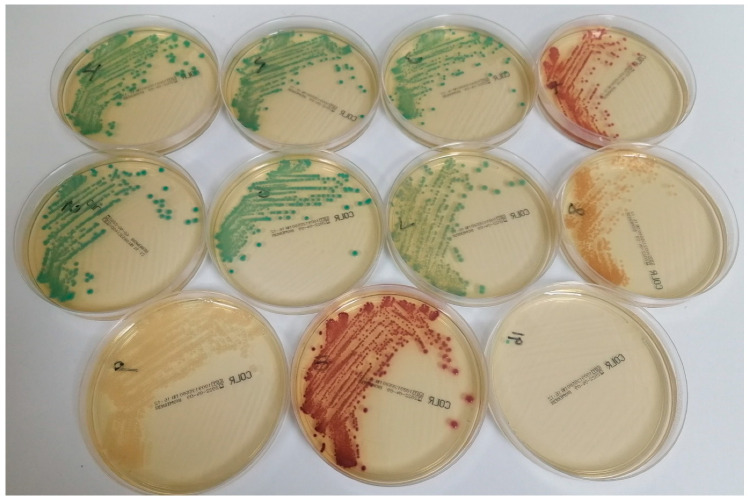
Appearance of various Gram-negative strains on CHROMID^®^ Colistin R after subculture of simulated blood cultures. The following species are shown (with colistin MICs in parenthesis): top row (left to right): *Klebsiella pneumoniae* (64 mg/L), *K. pneumoniae* (16 mg/L), *K. pneumoniae* (4 mg/L) and *E. coli* NCTC 13846 (4 mg/L); middle row (left to right): *K. pneumoniae* (32 mg/L), *K. pneumoniae* (32 mg/L), *E. cloacae* (8 mg/L) and *P. aeruginosa* (4 mg/L); bottom row (left to right): *Salmonella* species (8 mg/L), *E. coli* (8 mg/L) and *E. cloacae* (16 mg/L).

**Table 1 antibiotics-13-00246-t001:** Gram-negative species recovered from pre-selected blood cultures from patients in the UK (*n* = 326) and Türkiye (*n* = 64).

	No. of Isolates (*n*) from:	
	Newcastle upon Tyne (UK)	Ankara (Türkiye)
*Acinetobacter baumannii* complex	1	7
*Acinetobacter lwoffii*	2	-
*Acinetobacter* species	4	1
*Acinetobacter ursingii*	1	-
*Aeromonas* species	1	-
*Citrobacter freundii* complex	6	-
*Citrobacter koseri*	5	-
*Enterobacter cloacae* group	12	5
*Escherichia coli*	181	28
*Klebsiella oxytoca*	12	1
*Klebsiella pneumoniae* group	56	14
*Kluyvera cryocrescens*	1	-
*Mixta calida*	-	1
*Moraxella osloensis*	1	-
*Morganella morganii*	3	-
*Neisseria* species	1	-
*Pantoea* species	6	-
*Proteus mirabilis*	19	-
*Pseudomonas aeruginosa*	20	4
*Pseudomonas koreensis*	1	-
*Pseudomonas putida* group	3	1
*Pseudomonas* species	3	-
*Pseudomonas stutzeri*	2	-
*Raoultella* species	1	-
*Rhizobium radiobacter*	1	-
*Salmonella* Colindale	1	-
*Salmonella* Paratyphi A	1	-
*Salmonella* species	-	1
*Salmonella* Typhi	3	-
*Serratia marcescens* group	12	1
*Sphingobacterium* species	1	-
*Stenotrophomonas maltophilia*	2	2
*Superficieibacter electus*	1	-
Total	364	66

**Table 2 antibiotics-13-00246-t002:** Colistin-resistant isolates found among 443 Gram-negative isolates and their detection by CHROMID^®^ Colistin R.

	Simulated Blood Cultures (UK)	Detected	Real Blood Cultures (UK)	Detected	Real Blood Cultures (Türkiye)	Detected	Grand Total (All Samples)	Detected
*A. baumannii* complex	0	0	0	0	1	1	1	1
*Enterobacter cloacae* group	2	2	2	0	1	1	5	3
*Escherichia coli*	2	2	1	1	0	0	3	3
*Klebsiella pneumoniae* group	5	5	1	1	5	4	11	10
*Pseudomonas aeruginosa*	1	1	0	0	0	0	1	1
*Pseudomonas putida* group	0	0	1	0	0	0	1	0
*Salmonella* species	1	1	0	0	0	0	1	1
**Total (acquired resistance)**	**11**	**11**	**5**	**2**	**7**	**6**	**23**	**19**
*Morganella morganii*	0	0	3	3	0	0	3	3
*Proteus mirabilis*	0	0	19	19	0	0	19	19
*Serratia marcescens* group	0	0	11	11	1	1	12	12
**Total (intrinsic resistance)**	**0**	**0**	**33**	**33**	**0**	**0**	**34**	**34**
**Grand Total**	**11**	**11**	**38**	**35**	**8**	**7**	**57**	**53**

**Table 3 antibiotics-13-00246-t003:** Summary results for Enterobacterales from all blood cultures processed in the UK and Türkiye.

	Total (*n*)	Colistin Susceptible (*n*)	Growth on CHROMID^®^ Colistin R. (*n*)	Colistin Resistant *(n*)	Growth on CHROMID^®^ Colistin R. (*n*)
*Citrobacter freundii* complex	6	6	0	0	0
*Citrobacter koseri*	5	5	0	0	0
*Enterobacter cloacae* group	19	14	0	5	3
*Escherichia coli*	211	208	1	3	3
*Klebsiella oxytoca*	13	13	0	0	0
*Klebsiella pneumoniae* group	75	64	2	11	10
*Kluyvera cryocrescens*	1	1	0	0	0
*Mixta calida*	1	1	0	0	0
*Morganella morganii*	3	0	0	3	3
*Pantoea* species	6	6	0	0	0
*Proteus mirabilis*	19	0	0	19	19
*Raoultella* species	1	1	0	0	0
*Salmonella* species	7	6	0	1	1
*Serratia marcescens* group	13	1	0	12	12
*Superficieibacter electus*	1	1	0	0	0
Total Enterobacterales	381	327	3	54	51

**Table 4 antibiotics-13-00246-t004:** Summary statistics for CHROMID^®^ Colistin R for (i) Gram-negative bacteria and (ii) Enterobacterales only.

Total isolates considered	434		
True-positives	53	Sensitivity (%)	93.0
False-positives	9	Specificity (%)	97.6
False-negatives	4	Positive predictive value (%)	85.5
True-negatives	368	Negative predictive value (%)	98.9
Total Enterobacterales	381		
True-positives	51	Sensitivity (%)	94.4
False-positives	3	Specificity (%)	99.1
False-negatives	3	Positive predictive value (%)	94.4
True-negatives	324	Negative predictive value (%)	99.1

## Data Availability

Any unpublished raw data associated with this study are freely available by contacting the corresponding author.

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
