# Peer review of "The Use of CHROMID® Colistin R for the Detection of Colistin-Resistant Gram-Negative Bacteria in Positive Blood Cultures"

_antibiotics, 2024, doi:10.3390/antibiotics13030246_

Round 1

Reviewer 1 Report

Comments and Suggestions for Authors

This is a good contribution that should be helpful for other researchers.

The manuscript addresses an important obstacle in obtaining reliable results from tests for colistin resistance in clinical laboratories. The most reliable standard method broth dilution is cumbersome to perform, and standard plate based tests are not reliable. Therefore, the present approach of trying a new version of plate assay is welcome. The work is well planned and performed and the description is easy to follow. It should thus be useful for clinical bacteriology to add to the standard tests used. Even if it is not 100% reliable it is still better than other similar methods.

Author Response

Author response: We are grateful for your positive comments.

Reviewer 2 Report

Comments and Suggestions for Authors

Dear authors,

The research presented is very important to AMR diagnosis. The paper presenting good results. Thus, I have some suggestions and questions that I hope to improve your work.

- Do you have any information about the use of CHROMID colistin R in hospitals in the europe? or other countries? This information is very important to your discussion and to the aim of the paper.

- Table 4: Did you find any relationship between the MIC and the CHROMID COLR detection? Here is important the information about the colistin concentration in the Chromid Agar. Did you do any comparsion about the MIC and the detection of Klebsiella COLR and Enterobacter COLR? This information is very important as false negatives can complicate the infection treatment.

Discussion/Conclusion: What is your recommendation to the false negative situation? Do you recommended a MIC confirmation to negative CROMID COLOR (to avoid complication in the treatment)?

Author Response

Responses to reviewer 2

Dear authors,

The research presented is very important to AMR diagnosis. The paper presenting good results. Thus, I have some suggestions and questions that I hope to improve your work.

- Do you have any information about the use of CHROMID colistin R in hospitals in the europe? or other countries? This information is very important to your discussion and to the aim of the paper.

Author response: Unfortunately, this is commercially sensitive information that is not disclosed by the manufacturer. There is no published information on the scale of use of this medium.

- Table 4: Did you find any relationship between the MIC and the CHROMID COLR detection? Here is important the information about the colistin concentration in the Chromid Agar. Did you do any comparsion about the MIC and the detection of Klebsiella COLR and Enterobacter COLR? This information is very important as false negatives can complicate the infection treatment.

Author response: There was generally a good correlation between MIC and growth on CHROMID COLR since the vast majority of susceptible isolates were inhibited and the majority of resistant isolates were able to grow. We feel that this is already clear from the data without adding additional text. For the colistin resistant Klebsiella that failed to grow, there is no obvious explanation as it had a high MIC (> 8 mg/L). We have discussed how we believe the failure to detect colistin-resistant Enterobacter is due to the phenomenon of heteroresistance, as previously studied by others. The concentration of colistin in the medium is undisclosed by the manufacturer.

Discussion/Conclusion: What is your recommendation to the false negative situation? Do you recommended a MIC confirmation to negative CROMID COLOR (to avoid complication in the treatment)?

Author response: A good question. Our view is that further studies in different geographical locations are required to prove the accuracy of the method. Until the results of such studies are available it would be wise to be cautious. We have written in line 323 that “ If colistin therapy is under consideration for strains that appear susceptible, confirmation of the susceptible result using BMD is advisable, particularly for strains of Enterobacter species, at least until further studies are able to confirm the accuracy of this method.”

Reviewer 3 Report

Comments and Suggestions for Authors

In my opinion, this is an interesting short report that demonstrates the usefulness of screening blood culture isolates for colistin resistance using a simple method of streaking an aliquot directly from the BC bottle onto a chromogenic Col R agar to reduce time and workload. Although colistin is no longer recommended for the treatment of CPO, it may have some practical relevance when treatment options are limited for various reasons.

As the paper is based on a short evaluation of a very simple method, it should be changed from an article to a short communication. The results are convincing.

Tables 1, 3 and 4 should be combined into one table. Table 3 can be omitted. The numbers and species of gram-positive bacteria from clinical blood cultures should be reported in the text, as they are only negative controls.

Lines 129-132: In order to make a statement about the different prevalence of colistin resistance in Turkey and the UK, all BCs should have been systematically tested at both sites within the same time period of the project. I suspect that this was not the case.

Tables 6 and 7 should be combined.

Discussion: I suggest starting with the advantage of the method used here and then going on to compare the method and results with other studies. Please do not repeat results that have already been explained (lines 200-204).

Author Response

Responses to reviewer 3

In my opinion, this is an interesting short report that demonstrates the usefulness of screening blood culture isolates for colistin resistance using a simple method of streaking an aliquot directly from the BC bottle onto a chromogenic Col R agar to reduce time and workload. Although colistin is no longer recommended for the treatment of CPO, it may have some practical relevance when treatment options are limited for various reasons.

As the paper is based on a short evaluation of a very simple method, it should be changed from an article to a short communication. The results are convincing.

Author response: Thank you for your positive comments. After consideration of all of the reviews the assistant editor has requested that we increase the number of references so that manuscript is more consistent with an article rather than a short communication.

Tables 1, 3 and 4 should be combined into one table. Table 3 can be omitted. The numbers and species of gram-positive bacteria from clinical blood cultures should be reported in the text, as they are only negative controls.

Author response: Thank you for this useful recommendation. We have managed to combine tables 1 and 3 without any loss of clarity but it has not proved practical to also combine these with table 4. The reviewer has suggested deletion of table 3 – we assume they are referring to Table 2, which contained data for Gram-positives. This has now been deleted and replaced with text at the end of section 2.2.

Lines 129-132: In order to make a statement about the different prevalence of colistin resistance in Turkey and the UK, all BCs should have been systematically tested at both sites within the same time period of the project. I suspect that this was not the case.

Author response: We accept this point to a degree but (i) the studies were performed in the same year, (ii) the statistical difference is very strong and (iii) the high frequency of colistin resistance in Türkiye  - especially in Klebsiella  - is supported by many literature reports. Nevertheless we have added a caveat to state that “…..although the study in Türkiye was performed several months after the UK study”. (line 118).

Tables 6 and 7 should be combined.

Author response: These have been combined as requested and the tables renumbered. The number of tables has been reduced from 7 to 4.

Discussion: I suggest starting with the advantage of the method used here and then going on to compare the method and results with other studies. Please do not repeat results that have already been explained (lines 200-204).

Author response: The preference of the authors is to outline the most relevant literature to provide context and then discuss the results of our study so that our findings can be placed in that context. We have tried to minimise repetition of results by omitting two lines of text.